# The Mediating Role of Job Competence between Safety Participation and Behavioral Compliance

**DOI:** 10.3390/ijerph18115783

**Published:** 2021-05-27

**Authors:** Jia-Ming Wang, Pin-Chao Liao, Guan-Biao Yu

**Affiliations:** 1School of Economics and Management, Tongji University, Shanghai 200092, China; 1551288@tongji.edu.cn; 2Department of Construction Management, Tsinghua University, Beijing 100084, China; guanbiaoyu@163.com

**Keywords:** behavioral compliance, job competence, mediate, safety participation

## Abstract

The effective improvement of employee behavioral compliance and safety performance is an important subject related to the sustainable development of the construction industry. Based on data from a Chinese company (*n* = 290), this study used a partial least squares-structural equation model to clarify the relationship among safety participation, job competence, and behavioral compliance. Empirical analysis found that: (1) safety participation had a significant positive impact on employees’ behavioral compliance; and (2) job competence played a partial mediating role between safety participation and behavioral compliance. By selecting two new perspectives of safety participation and job competence, this study derived new factors affecting behavioral compliance, constructed a new theory about safety management, and conducted an in-depth discussion on improving behavioral compliance theoretically. Practically, the research put forward a new decision-making model, deconstructed the mechanism between safety participation and behavioral compliance, and provided new guiding strategies for improving employee behavioral compliance.

## 1. Introduction

Safety behavioral compliance is usually defined as “the core safety activities that need to be carried out by individuals to maintain workplace safety” [1]. At the construction site, the lack of safety behavioral compliance poses safety risks and a higher probability of accidents among workers [2]. In the construction industry, hundreds of thousands of people die every year due to safety accidents. Considering the United States as an example, according to official statistics, the number of deaths in the construction industry nationwide in 2018 was 1038 [3]. The death rate was significantly higher than that of any other industry, and there has been no obvious downward trend in recent years [3]. Falls, electric shocks, and impacts caused by violations of safety regulations are among the main causes of death [4]. Therefore, in the context of continuous severe safety issues in the global construction industry, the effective improvement of worker safety behavioral compliance and ensuring worker safety has become major strategic concerns related to the sustainable development of the construction industry [5].

To improve workers’ safety behavioral compliance, scholars have conducted certain research. From research perspectives, scholars have studied behavioral compliance issues mainly from the perspectives of employees, employers, and regulatory agencies [6,7,8]. In terms of research content, scholars have mainly studied the impact of three levels of factors—personal characteristics of employees, manager characteristics, and organizational characteristics—on the safety behavioral compliance [9,10]. From the perspective of research methods, there are more qualitative research methods in the literature related to improving employees’ behavioral compliance, and a few studies have used expert scoring methods for simple quantitative analysis [11,12]. It can be observed that, theoretically, the current research on improving safety compliance is mainly at the stage of qualitative analysis, and the influencing factors of the research are relatively monotonous. As a result, new influencing factors need to be introduced to enrich the safety management theory. In practice, project managers still have considerable confusion in improving employee safety behavioral compliance, which implies a lack of more specific targeted measures in management practices. Therefore, this research has both practical and theoretical needs.

Studies have shown that in safety accidents caused by operating behaviors that do not comply with safety regulations, the lack of worker participation in safety activities is an important cause for operating violations [13]. In project management, participation in safety-related activities is usually expressed through the concept of “safety participation.” The theory of social norms specifies that a “norm-oriented organization” usually refers to behaviors that are generally considered acceptable and appropriate, and the behavior of individuals in the organization will be affected by the organizational environment constructed by norm-orientation [14]. Neal and Griffin’s definition of safety participation is “behaviors that do not directly contribute to an individual’s personal safety but that do help to develop an environment that supports safety” [15]; because the safety participation of employees belongs to the category of individuals integrating into the organizational environment, safety participation also has a significant impact on the compliance of employee behavior. However, the existing literature such as the safety performance framework proposed by Neal and Griffin divides “safety behavior” into two parallel sub-dimensions of “safety participation” and “safety compliance,” ignoring safety participation’s possible impact on behavioral compliance [16]. Later scholars have also conducted research on the basis of this paradigm, and a large number of studies have researched safety participation as an explained variable, rather than as an explanatory variable [17,18]. As “safety participation” is relatively easy to operate compared with other factors, influencing behavioral compliance and ignorance of this concept has led to an imperfection in safety management theory and the confusion of managers in practice.

In order to explore the impact of safety participation on behavioral compliance, it is necessary to clarify the mechanism between safety participation and behavioral compliance. According to the theory of social learning, human development is the result of the interaction among human beings, environment, and society. Environment and society can influence human behavior, while humans can also acquire abilities by observing the environment and society [19]. Therefore, employees, through continuous observation and learning in safety participation, are likely to enhance their own abilities at work, and the improvement of employees’ abilities may promote the improvement in safety behavior compliance. In the field of management, the ability of employees at work is usually expressed by job competence. However, research on work competence is currently mainly focused on the evaluation of the teachers’ job competency, the improvement in the college students’ job competence before employment, the estimation of the medical staff’s job competency, and so on [20,21,22]. Few people have paid enough attention to the job competence of employees in engineering projects. This article defines job competency as “the ability to perform specific tasks and roles within one’s own organization” [23]. This concept may be a key mechanism for explaining the relationship between safety participation and behavioral compliance.

Therefore, this research raises the following questions: (1) What impact does safety participation have on behavioral compliance? (2) What role does job competence play between safety participation and behavioral compliance? (3) From this perspective, how should project managers improve their work?

As a consequence, this study aimed to explore the specific relationship between job competence, safety participation, and compliance behavior. The main research steps were as follows. (1) First, this paper proposed a mediation model through the literature review and theoretical derivation; (2) Next, this paper conducted a structured survey on the full-time employees of China’s Sichuan Petrochemical Company and tested the research hypothesis; and (3) Finally, initiating from the results, this article conducted an in-depth discussion from both theoretical and practical perspectives.

The contribution of this research is divided into four aspects: (1) breaking through a shallow qualitative analysis, we used empirical analysis to quantitatively study the relationship between safety participation, job competence, and behavioral compliance; (2) deriving new factors influencing safety behavioral compliance, we provided a new perspective for safety behavioral compliance research; (3) a new decision-making model and a new theory in the field of behavioral compliance were constructed; and (4) in practice, to improve employee behavioral compliance and build a safety evaluation standard in engineering projects, we created new guiding strategies that are helpful to the management practice of large-scale engineering projects.

## 2. Literature Review and Hypotheses

### 2.1. The Impact of Safety Participation on Compliance Behavior

As the main standard for measuring safety behaviors, safety participation generally includes safety training programs and other activities that organizations, employers, and employees participate in to improve workplace safety [24]. Scholars such as Ghani considered that in the field of construction engineering, effective safety training is of great importance to improve employee safety awareness and modify employee behavior [25]. Since then, many scholars have put forward their views through qualitative analysis. Sulastre Mat Zin et al. conducted a qualitative analysis about safety issues in the construction industry and argued that participation in safety training and other activities was an important factor affecting the compliance of employee behavior [26]. Zin et al. further expanded the field of this view, believing that effective safety training activities in which employees have participated can be transformed into behavioral compliance at work [27,28].

A few scholars have also used empirical analysis to conduct preliminary explorations. McDonald conducted a survey on 18 construction sites in Ireland and pointed out that employees’ insufficient participation in safety training was the root cause of construction site accidents because they did not have enough knowledge and skills to identify potential hazards at the construction site [29]. Fernando et al. conducted a similar study in the petrochemical field. Considering 169 employees in the petrochemical processing zone in northern Malaysia as a sample, empirical research showed that employees who participated in safety training generally had a higher level of safety behavioral compliance [30]. Hsinkuang Chi et al. introduced this point of view into the medical field. Based on the data collected from 732 samples of Vietnamese hospitals during the COVID-19 epidemic, they initially concluded that there was a correlation between safety training and behavioral compliance [31]. In addition, Vinodkumar and Bhasi found that if the organization did not regularly carry out safety activities, the safety compliance of employee behavior would decrease [32]. Based on the literature review above-mentioned, the following hypothesis was proposed:

**Hypothesis** **1** **(H1).**
*Safety participation has a significant positive impact on behavioral compliance.*


### 2.2. The Impact of Safety Participation on Job Competence

The term “competence” can be understood as a cluster of knowledge, skills, and abilities possessed by an individual for performing duties in an organization. In many studies about organizational safety management, job competence has been an important indicator for evaluating employee job performance [22]. The competency theory points out that knowledge and skills are the core of the job competence of organization members, and the acquisition of knowledge and skills is closely related to the input of effective information in the environment [33,34]. As employee safety participation is a process of effectively interpreting information from the environment, employees may continue to observe and learn through safety participation, and ultimately enhance their safety knowledge and safety skills.

Research in multiple industries in recent years has shown that there may be a significant relationship between safety participation and work competence. Robert et al. made an empirical analysis with a sample of 402 employees from 31 restaurants in three Midwestern states and found that workers participating in safety-related programs generally improved their personal food safety protection knowledge [35,36]. The study of Salleh et al. discovered that in the chemical engineering industry, the lack of employee participation in safety training programs could lead to insufficient job competence [37]. Yue Shen et al. investigated many hospitals in China and demonstrated that among the same batch of newly recruited nurses, the nurses who had experienced psychological safety training had better job competence [38]. In addition, HSE also recommended that employees were supposed to participate in a series of safety-related training programs (such as core training activities) to improve job competence [39,40], which reflected the significant impact of safety participation on job competence. Therefore, we proposed the following hypothesis:

**Hypothesis** **2** **(H2).**
*Safety participation has a significant positive impact on job competence.*


### 2.3. The Impact of Job Competence on Behavioral Compliance

From the perspective of agency theory, the top executives of a company can be regarded as principals, and frontline employees can be regarded as agents [27]. As shown in previous research, when the principal requires the agents to perform certain specific behaviors, and these specific behaviors are beneficial to the principal but costly to the agents, problems may arise regarding the agents’ behavioral compliance [28]. When faced with the same behavioral compliance requirements, employees with higher job competence often only need to make less effort with regard to time, energy, physical fitness, etc. to meet the compliance requirements [41]. This suggests that the improvement of job competence can reduce the cost of employees to make required behaviors, and hence, job competence is likely to significantly improve the level of safety behavioral compliance.

At present, scholars have been involved in preliminary discussions on the issues of job competence and safety. Through qualitative analysis, some scholars believe that safety knowledge and safety skills are important factors affecting employee behavioral compliance. Neal and Griffin’s survey showed that the employees’ levels of knowledge and skill were positively correlated with the level of safety compliance behavior [42]. Some scholars have studied this from the alternate view. Fairman and Yapp have proven through quantitative research that employees’ lack of knowledge and skills is an important cause of safety violations [43]. Additionally, research in the medical field also supports similar views. Tae and Hwang demonstrated that the clinical competence of nurses was significantly positively correlated with the level of safety compliance behavior and proposed that education and training were essential means to improve the level of safety compliance behavior of nurses [44].

Accordingly, we proposed hypothesis 3:

**Hypothesis** **3** **(H3).**
*Job competence has a significant positive impact on behavioral compliance.*


### 2.4. The Mediating Role of Work Competence

According to the discussion in Section 2.2 and Section 2.3, we believe that safety participation can not only directly affect behavioral compliance, but can also indirectly affect behavioral compliance through the “bridge” of job competence. On one hand, existing studies have shown that safety participation with vocational education and training could provide employees with professional skills and knowledge, thereby improving their work capabilities. Therefore, safety participation can be regarded as a “platform” for employees to learn work-related knowledge and skills to improve their job competence [34,35]. On the other hand, some scholars have found that an improvement in employee job competence would make employees more adaptable to work and ultimately improve the level of their own safety compliance behavior; that is, job competence has a significant impact on the compliance of employee behavior [41,42]. In summary, safety participation first has a positive impact on job competence, and subsequently, job competence has a positive impact on behavioral compliance. Thus, we proposed the following hypothesis:

**Hypothesis** **4** **(H4).**
*Job competence plays a mediating role in the relationship between safety participation and behavioral compliance.*


Accordingly, we proposed the conceptual model presented in Figure 1.

## 3. Methods

### 3.1. Data Collection

The Department of Construction Engineering Management of Tsinghua University performed this study as an employer to study the mediating effect of job competence between safety participation and behavioral compliance. This paper combined qualitative analysis with quantitative research to explore the relationship between safety participation, work competence, and behavioral compliance. After completing the qualitative analysis through a theoretical derivation and a review of the literature, we used structured interviews to conduct surveys among a sample of employees from an infrastructure construction company in Sichuan, China (PCSP).

Above all, our experiment was approved by the institutional ethics committee (project code, THU201914), and we signed a legal agreement with the study participants, guaranteeing not to collect their highly sensitive information (such as mobile phone numbers, credit card information, and various passwords). Ethics approval documents mainly included research background, research goals, methods and research materials, recruitment and randomization of subjects, research procedures, possible risks or harm to participants caused by the research, withdrawal or suspension of the research, privacy, and confidentiality. Informed consent was obtained from all the participants involved in the study, and the study was conducted according to the guidelines of the Declaration of Helsinki. Moreover, we ensured that all data were fully anonymized and carefully restored to protect the sample’s privacy.

We decided to adopt structured interviews in this study based on the following reasons. First, this research intended to use quantitative methods, and the results of structured interviews can be easily quantified. Second, structured interviews allow for the reliability of the survey results to be assessed easily and offer the advantage of a high recovery rate, which is important for the process of sample collection. Third, because this research is relatively complex, and structured interviews are suitable for asking complex questions, choosing structured interviews can help us conduct in-depth investigations on safety-related issues. Finally, structured interviews enable us to observe the interviewee’s attitude and behavior including non-verbal information that cannot be obtained by just filling in a written questionnaire.

In order to make this research more authoritative, we hired a professional team from Beijing PetroChina Oriental Integrity Certification Consultant Co. Ltd. (BPOS) to conduct the structured interviews. BPOS is an authoritative consulting organization, formally approved by the China National Certification Supervision and Administration Commission, which helped us to ensure the scientificity and rationality of the structured interviews. Based on the nature of risks and job responsibilities of each department, the research team formulated the content of the corresponding interview list for indicators such as job competence and paid attention to the flexibility of the questions to make sure that the assessment was fair and reasonable. In order to confirm the authenticity of the interviewees’ expressions, the researchers compared every interviewee’s answer with relevant documents and materials during the interview. In addition, the evaluation experts were divided into two groups to ensure the objectivity and fairness of the results. After the interview, the two groups of evaluation experts compared and discussed the scores until they reached an agreement.

The total sample collected in this study from the infrastructure construction company in Sichuan was 291, and the final sample size after removing invalid values and outliers was 290. Table 1 shows the basic information of the sample. Moreover, 96.2% of the sample were male, while 3.8% were female. Furthermore, the age of 74.1% of the workers ranged from 31 to 39 years. The analysis also showed that 46% of the workers were from the fourth production sector and 54% were from the sixth production sector. In terms of work positions, 7.6% were leaders, 40.3% were external operators, and 52.1% were internal operators. Finally, 77.2% of the workers had been working in the company for a period of six to 10 years.

The data obtained in this study had a certain degree of sensitivity. Ensuring data security, ethics, and personal privacy are issues of great importance. We achieved this goal mainly through the following measures. Before data collection, we obtained ethical approval and signed an ethical agreement with the study participants. Subsequently, we communicated with them thoroughly, ensuring that they were fully informed. During data collection, unrelated personnel were strictly prohibited from entering and taking photos at the test site to prevent data theft. Researchers could only bring the allowed communication and filming equipment into the test site. After data collection, we anonymized all the information and then started the analysis to protect privacy effectively. Our research team deleted all name-containing data. In addition, we did not collect highly sensitive private information such as mobile phone numbers, credit card information, and various passwords. After the analysis was completed, the data were stored on a safe laboratory computer’s hard drive, thus effectively preventing the data from being stolen. Moreover, we deleted all privacy-related data from the computer used for data analysis and shredded the relevant files to make them unrecoverable.

In fact, the sample in this article was highly representative. On one hand, the engineering project from which we selected samples was highly representative. The project was a comprehensive power station construction site located in the city of Nanchong, Sichuan Province, China. There were a large number of complicated buildings under construction, which was a typical construction project. As the project had a complex and complete construction process, the company needed to be equipped with a full range of technical work types. On the other hand, the individual samples themselves were highly representative. The research team detailed the basic information of the samples including name, gender, age, department, work experience, position, marital status, family composition, income level, education level, etc. However, due to privacy considerations, data such as name, marital status, family composition, and income level could not be provided. It can be concluded from the descriptive statistics that the sample was composed of male grassroots employees with an average age of about 33.94, and there were a small number of women and managers. This is basically similar to the composition of construction workers in most countries in the world. In summary, the samples were broadly representative.

### 3.2. Measures

To conclude, the variables measured in this study mainly included behavioral compliance, job competence, and safety participation. The specific measurement methods for the variables are listed in Table 2.

#### 3.2.1. Behavioral Compliance (BC)

Behavioral compliance was measured by adopting two suitable items from Griffin and Neal’s model [1]. One of these items, “I use and wear appropriate personal protective equipment at work”, was measured on a 5-point scale, with 1 representing I always forget to use the correct personal protective equipment, and 5 representing I strictly use the correct personal protective equipment. The other item was “I follow all kinds of safety regulations and procedures at work.” Likewise, a 5-point scale was used, with 1 meaning I rarely comply with safety regulations and procedures, and 5 meaning I strictly comply with safety regulations and procedures.

#### 3.2.2. Job Competence (JC)

In order to measure the job competence for each worker, we specifically adopted four items used by BPOS for assessing the worker’s knowledge and skills. All items were scored on a 5-point scale, with 1 being basically clear, achieving acceptable results, and 5 being fully clear, completely achieving the expected result, or beyond the expected result. The items inquired about the familiarity with the equipment, facilities, tools, and instruments in the work position, the understanding of the existing risks, maintenance, and relevant safety requirements, ability to operate tools correctly, familiarity with the emergency procedures and responsibilities, the use of positive-pressure respirator, antifreeze clothing, fire clothing, gas mask, fire extinguisher, other emergency equipment, and so on.

#### 3.2.3. Safety Participation (SP)

In order to measure safety participation, three items were used to assess the extent to which individuals participated in safety-related activities. For instance, items included “I participate in on-post safety inspection and join the working team in routine on-site inspection” and “I participate in the activities of hazard identification and risk assessment.” All items were scored on a 5-point scale, with 1 representing “I never attend a safety-related activity” and 5 representing “I participate in every safety-related activity.”

#### 3.2.4. Control Variables

We selected two control variables, age and working experience, to reduce the effect of the different size of age groups. We regarded working experience as a continuous variable and measured it according to years and age. The mean age of the participants was 33.94, with a standard deviation of 5.09, and the mean working experience of the participants was 13, with a standard deviation of 8.40.

### 3.3. Analytical Approach

Compared with conventional regression analysis methods (such as linear regression), the structural equation model (SEM) has more flexibility, as it can test multiple equations at the same time [45]. Hence, SEM is more suitable for this study. SEM can be mainly divided into two types: covariance-based-SEM (CB-SEM) and partial-least-squares-SEM (PLS-SEM). PLS-SEM can not only process various structures and indicators for researchers, but the results are also more accurate than those obtained with other forms of SEM when the model has more than two variables [46]. In addition, PLS-SEM can better meet data processing requirements and adjust the relationship among variables [46]. Therefore, PLS-SEM was considered the most suitable analysis method for this study.

In this study, SmartPls 3.0 software, developed by SmartPLS located in Bönningstedt, Germany, was selected as the analysis tool. SmartPls has features that can simplify the analysis and is highly accessible. Furthermore, the user interface is easy to operate [47]. As a result, this study drew on the method of Zhao et al. and used the PLS-SEM model to conduct an empirical analysis [48].

## 4. Results

### 4.1. Measurement Model Evaluation

Since this model is fully reflective, it is necessary to thoroughly evaluate indicator reliability, internal consistency reliability, convergent validity, and discriminant validity [49]. We assessed all the criteria above-mentioned as follows.

#### 4.1.1. Indicator Reliability

In order to measure the reliability of each indicator, the factor loading should be examined. According to Hulland, a threshold value of 0.7 or higher for each indicator’s loading is considered as reliable. As can be observed in Table 3, the values obtained were higher than 0.7; thus, the outer loadings indicate an acceptable level of reliability [50].

#### 4.1.2. Internal Consistency Reliability

To verify the internal consistency reliability, the first step is the use of composite reliability to evaluate the model. Traditionally, researchers [51] have used Cronbach’s alpha. However, recently, Hair et al. suggested that composite reliability could provide a more appropriate way to measure internal consistency reliability for at least two reasons [52]:(1)Unlike Cronbach’s alpha, composite reliability does not need to assume that all indicator loadings are equal in the population, which is in line with the working principle of the PLS-SEM algorithm that prioritizes the indicators based on their individual reliabilities.(2)Cronbach’s alpha is also sensitive to the number of items in the scale and generally tends to underestimate the internal consistency reliability.

Consequently, by using composite reliability, PLS-SEM is able to accommodate different indicator reliabilities, which effectively avoids the underestimation associated with Cronbach’s alpha. Hair et al. assumed that the composite reliability should be equal to or higher than 0.7 to satisfy the standard of internal consistency reliability [53]. As shown in Table 3, such values were higher than 0.7, ranging from 0.810 to 0.903. Accordingly, high levels of internal consistency reliability were demonstrated.

#### 4.1.3. Initial Validity Test

After data collection, we anonymized the samples to protect privacy. Subsequently, we used SPSS23.0 to test the initial validity of the questionnaire. When using factor analysis for validity testing, some prerequisites should be satisfied—there should be a suitable correlation between the measured items, as reflected by the Kaiser–Meyer–Olkin (KMO) value and Bartlett’s sphericity test value. Among them, KMO, whose value is within the 0–1 range, was used to compare the partial correlation coefficients as well as simple correlation between the items. The standards for the indicator were: greater than 0.9 (totally suitable), 0.7–0.9 (very suitable), 0.6–0.7 (suitable), 0.5–0.6 (not suitable), and 0.5 and below (the data are not supposed to be used for further study). Bartlett’s sphericity test value was used to decide whether the correlation coefficient between the items was significant or not. If the value was significant (sig. < 0.05), it was suitable for the factor analysis.

Finally, Table 3 shows that the minimum of KMO value was 0.812, which is in the 0.7–0.9 range, showing that the scale in this questionnaire was very suitable for further research. As for Bartlett’s sphericity test results, the chi-squared value was 249.881, which was high and proved that the corresponding *p*-value was <0.05. Hence, Bartlett’s sphericity test was significant.

To conclude, the initial validity of the research showed that the data were suitable for this empirical study.

#### 4.1.4. Convergent Validity

Bagozzi and Yi recommended the use of average variance extracted (AVE) to examine convergent validity [54]. Furthermore, an AVE value of 0.5 or higher indicates a satisfying degree of convergent validity, implying that the latent variable explains more than half of its indicators’ variance. Again, as shown in Table 4, all of the values of AVE were above the acceptable threshold of 0.5, thus confirming convergent validity.

#### 4.1.5. Discriminant Validity

In order to assess the discriminant validity, the Fornell–Larcker criterion and cross loadings need to be examined. Fornell and Larcker suggested that the square root of AVE in each latent variable could be used to assess discriminant validity [55]. If its value is higher than all other correlation values among the latent variables shown in Table 5, the discriminant validity could be effective.

For cross loadings, Chin proposed that each indicator loading should be greater than all of its cross loadings [56]. As can be observed in Table 6, the cross loadings criterion was fulfilled. Thus, the result fully confirms the discriminant validity.

### 4.2. Structural Model Evaluation

To use the PLS-SEM model, we followed the general recommendations of Hair and colleagues to test the mediating effects [57]. First, we measured the R^2^ to observe the explanatory power of the model. Second, the bootstrapping technique was adopted to assess the path coefficient’s significance by randomly selecting cases with replacement from the original sample. Third, we used blindfolding to achieve cross-validated redundancy measures for each construct (predictive relevance: Q2).

The aforementioned techniques were considered in this study because PLS-SEM had a greater statistical power compared with CB-SEM, and it is more likely to identify significant direct effects [58]. Consequently, we assessed the direct relationship between safety participation and behavioral compliance without the mediator variable job competence to test hypothesis 1. Figure 2 represents the results of the direct effect model.

As shown in Figure 2, the value of R^2^ was 0.385 for behavioral compliance, which means that the latent variable safety participation explains 38.5% of the variance in behavioral compliance. Besides evaluating the magnitude of the R^2^ value as a criterion of predictive accuracy, researchers have suggested that it is also important to examine the Q2 value as another criterion of predictive relevance. A value of Q2 that is higher than zero for a certain endogenous latent variable indicates that the PLS path model has predictive relevance for the model. The Q2 value of latent variables in the PLS path model was obtained by following the blindfolding procedure in SmartPls 3.0 [53]. As shown in Figure 2, the Q2 value for behavioral compliance was 0.254, which exceeded zero, thereby confirming the predictive relevance of the constructed model. We also applied the standard bootstrapping procedure with a number of 5000 bootstrap samples and 290 cases to assess the significance of the path coefficient [53]. When the model was estimated without a mediator variable, the analysis (Figure 1) revealed a significant positive relationship between safety participation and behavioral compliance (β = 0.621, *p* < 0.01), supporting hypothesis 1.

Next, we assessed the model with job competence as a mediating variable. As shown in Figure 3 and Table 5, the R^2^ for behavioral compliance was 0.452, indicating that safety participation and job competence were able to explain 45.2% of the variance in behavioral compliance. By running the blindfolding procedure, the Q2 values for job competence and behavioral compliance were 0.212 and 0.297, respectively, proving the predictive relevance of the model.

Moreover, we also applied the standard bootstrapping procedure with 5000 bootstrap samples and 290 cases to assess the significance of the path coefficients with job competence in the model. As can be observed in Table 5, there was a significant positive correlation between safety participation and job competence (β = 0.571, *p* < 0.01), which supports hypothesis 2. Meanwhile, a significant positive correlation between job competence and behavioral compliance was also demonstrated (β = 0.317, *p* < 0.01), supporting hypothesis 3. A clear mediation effect in the model was indicated by the fact that the indirect path was significant (β = 0.181, *p* < 0.01). In addition, the direct path was also significant (β = 0.439, *p* < 0.01), and hence, the type of mediation can be considered as partial mediation with a positive path coefficient (0.317 * 0.571 ≈ 0.181) [48]. In other words, job competence partially (and complementary) mediates the relationship between safety participation and behavioral compliance, thus supporting hypothesis 2.

Since mediation analysis regularly involves partial mediation, it is helpful to have further information on the mediated portion [57]. Therefore, it is necessary to find out the mediating strength (the ratio of the indirect-to-total effect), which is known as the variance accounted for (VAF) value. Hair et al. suggested that VAF >80% indicates full mediation, 20% ≤ VAF ≤ 80% indicates partial mediation, and VAF <20% indicates no mediation [58].

As shown in Table 7, the portion of partial mediation had a VAF value of 31.7%, and therefore, job competence mediated 31.7% of the relationship between safety participation and behavioral compliance. We assume that there are several reasons regarding why the VAF value was 31.7%. For example, job competence may be one of the possible mediating variables between safety participation and behavioral compliance. It is, therefore, necessary to explore more mediating variables in the future. It should also be considered that there may be other factors affecting compliance behavior in addition to job competence, safety compliance, and control variables.

In summary, the four hypotheses of this study were verified. After repeated inspections, the results obtained from each analysis were consistent, indicating the reliability and validity of this study.

As for the common method bias, compared with traditional linear regression, SEM is more flexible, with a higher accuracy. This model is able to test multiple equations at the same time, allowing independent variables and dependent variables to contain measurement errors, and estimate the factor structure, factor relationship, and the entire model fit. As a result, this method has very little bias. After meticulous checking, it was observed that this model’s GFI was 0.932 (>0.9), RMSEA was 0.034 (<0.08), and CFI was 0.954 (>0.9), which demonstrates a good model fit. In conclusion, the common method bias is very low.

## 5. Discussion

The results of this research have been forwarded to two associate professors in the Department of Construction Management of Tsinghua University. The two experts agreed with the research conclusions that safety participation and behavioral compliance are significantly and positively correlated and that job competence plays a partial mediating role in the relationship between safety participation and behavioral compliance. The two experts believe that this research has important theoretical and practical significance in the context of the global construction industry, which regularly faces severe safety issues.

### 5.1. Theoretical Contribution

First, we would like to discuss the theoretical contribution of this research.

In terms of behavioral compliance, existing research usually focuses on medical engineering, food safety engineering, traffic safety engineering, and other fields. Margaret Ndagire and others used a questionnaire survey to analyze the behavioral compliance of motorcyclists in Kampala, Uganda, and came up with key factors to improve motorcyclists’ safety performance [59]. Scholars such as Hsinkuang Chi conducted surveys in Vietnamese hospitals during the COVID-19 pandemic and analyzed the effects of safety training on behavioral compliance [32]. Regarding the influencing factors of behavioral compliance, however, researchers have mostly conducted research from the organizational environment level and the managerial level, lacking in attention to the individual level of employees, especially focusing on the exploration of variables such as safety culture, safety atmosphere, and safety environment. On one hand, the theoretical contribution of this research lies in breaking through shallow qualitative analysis and using empirical analysis to quantitatively study the relationship between safety participation, work competence, and behavioral compliance. On the other hand, this research introduces behavioral compliance into the construction field and puts forward factors that affect behavioral compliance from the employee level, providing a new perspective for research in safety behavioral compliance.

As far as safety participation is concerned, existing studies have mostly analyzed safety participation as an explained variable. For example, Wei Wei et al. studied the impact of work–family conflicts and job satisfaction on drivers’ safety participation [18]. Lixin Jiang et al. explored the impact of transformational organizational leadership, safety knowledge, and safety motivation on safety participation from the perspective of corporate management [60]. However, few reports have considered the relationship between safety participation and job competence. This study confirmed not only the significant positive impact of security participation on behavioral compliance through empirical analysis, but also the significant positive impact of safety participation on job competence. Our results revealed a new path for safety participation to affect behavioral compliance, established a new mechanism between safety participation and job competence, and enriched engineering safety management theory thereby. The findings of this research show that if employees actively participate in safety-related activities such as regular on-site inspection training, hazard identification, and risk assessment, they can improve their job competence and attain higher safety behavioral compliance. This finding also confirms the results of previous research showing that unstable participation in safety training weakens the employee’s motivation to obey safety regulations and increases the possibility of violations of safety regulations [61]. If there is no continuous training or professional education, employees will lack knowledge of the safety rules, and in actual operation, they will often perform their task purely based on experience, without knowledge of safety rules, which leads to a higher level of risk [30].

In terms of job competence, existing research mainly focuses on the relationship between job competence and organizational or individual job performance, mainly in fields such as medicine, education, and national defense. For example, Song Mi Kyeong et al. used Korean hospitals as a sample to explore the relationship among competency, job satisfaction, and the performance of medical staff [21] and Dolores L. Arteche et al. studied the competence and effectiveness of part-time job retirees in colleges and universities [22]. Current scholars have not conducted in-depth discussions in the field of engineering safety. This study not only found the influencing factor of job competence, but also confirmed the significant positive impact of work competence on behavioral compliance. Therefore, this research further expands the competence theory and introduces work competence into the field of engineering safety management, enriches the influencing factors of safety behavioral compliance, and provides new explanations for differences in compliance behaviors. Job competence includes a collection of success factors (knowledge and skills) necessary to finish tasks [62]. In contrast with previous studies, this study emphasizes the partial mediating role of competence, which also suggests a more specific way for organizations to improve safety performance. On one hand, organizations can determine the corresponding safety training content according to various indicators of job competence, so as to make the training content more accurate and avoid a waste of resources. Some scholars have proposed that competent employees should be able to grasp ways to deal with the various problems that may arise and solve them according to the specific situations. As a result, the organization needs to train employees to solve various problems that they may face [63]. On the other hand, research shows that when employees start to gain competence in a new task or develop competence to a new level, they often ignore their limitations, which means that employees are not aware of what they cannot do. This phenomenon hinders the improvement in safety compliance. Therefore, when organizing training activities, managers should not only let employees clarify what they can do, but also make them aware of the limitations of their abilities. This may allow improvements in organizational safety performance.

### 5.2. Practical Contribution

Based on the above theoretical analysis, in order to improve organizational safety performance, we put forward the following three practical suggestions for project safety management:(1)Attach importance to the role of work competence and improve safety participation and behavioral compliance based on job competence. The research conclusions of this paper show that job competence plays a partially mediating role between safety participation and behavioral compliance. On one hand, managers need to divide job responsibilities carefully, improve the job competency evaluation system for each job, and improve compliance behavior from the perspective of job competence. In this manner, they can effectively avoid improper employment due to unclear division of job competency indicators. On the other hand, managers need to fully understand the multidimensional nature of job competence, design training content according to the dimensions of job competence, and reduce the waste of organizational resources caused by ineffective training.(2)Prioritize the employees’ safety participation and accurately design a safety training system based on job competence requirements. This study confirms that safety participation has a significant positive impact on job competence. In order to ensure effective learning, safety training for employees should include oral presentations as well as on-site observations or practical operations. At the same time, during the training process, managers should not only help employees acquire new knowledge and skills, but enable them to clarify the limitations of their abilities and to put themselves in a position that allows them to avoid the loss of organizational safety performance caused by the overestimation or underestimation of job competence.(3)Comprehensively consider the two influencing factors of behavioral compliance, attach importance to the spill effect brought about by the improvement of behavioral compliance, and create a strong safety culture. This research shows that both job competence and safe participation can affect behavioral compliance. Consequently, in order to improve organizational safety performance, managers should comprehensively consider the factors influencing behavioral compliance. The improvement in organizational safety performance will generate spillover effects including the enhancement of the organizational safety culture. Competency theory points out that employee job competence and activity participation can be regulated by the corporate culture. As a result, in order to enhance the safety participation and job competence of employees, companies should enhance the safety culture through a series of safety culture building activities, especially, periodic brand cultural activities to promote the improvement in organizational safety performance.

Regarding the relevant recommendations of the HSE [23], we proposed a management model of job competence based on the PDCA (plan-do-check-act) management model and social norm theory [64]. As shown in Figure 4, after dividing the work roles, managers should first arrange positions for the employees and formulate the standards for job competence evaluation as well as for education and training programs [65]. Next, managers should organize the employees to participate in the corresponding education and training programs and conduct the assessment according to the job competence standards [66]. Afterward, job competence levels should be divided according to the results. Finally, employees who meet the standards should be allowed to work and participate in further training, while those who do not meet the standards should be retrained or dismissed [67].

## 6. Conclusions

This research aimed to explore the relationship between job competence, safety participation, and behavioral compliance. The empirical analysis found that: (1) safety participation had a significant positive impact on behavioral compliance, while (2) job competence played an intermediary role between safety participation and behavioral compliance. Theoretically, this research discovered a new path for safety participation to influence compliance behavior, constructed a new theoretical model of behavioral compliance, and enriched safety management theory for use in engineering projects. This research provides a practical basis for job competence and behavioral compliance evaluation standards and provides effective suggestions and guidance to improve the safety performance of large-scale projects.

Although this research has a unique value, it also has certain limitations, which provide future researchers with meaningful directions. First, restricted by current research conditions, the samples in this study were obtained mainly from China. Moreover, in a statistical sense, there are inevitably errors in any sample; therefore, future researchers should use samples of employees from different industries. Second, there may be an inverse causal relationship between safety participation and behavioral compliance, currently set as independent and dependent variables. Since our research team has been engaged in engineering safety research for a long time, in future research, we will consider measuring variables at multiple time points to better solve the problem of reverse causality. Although the new intermediary variable is beyond the scope of this article, it is worthy of further discussion. Finally, besides job competence and behavioral compliance, there may be other factors affecting behavioral compliance. As a result, more variables can be used in future studies to build a more complex model.

## Figures and Tables

**Figure 1 ijerph-18-05783-f001:**
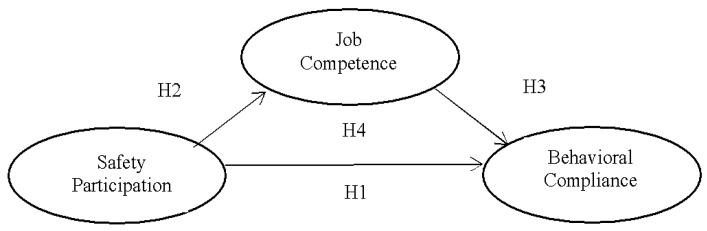
Theoretical model.

**Figure 2 ijerph-18-05783-f002:**
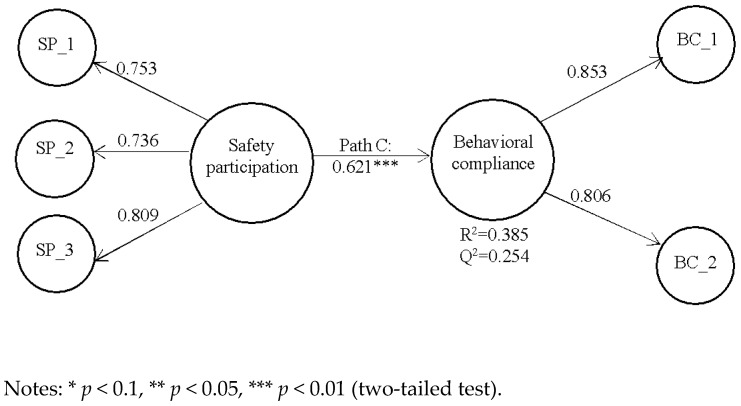
The direct effect model.

**Figure 3 ijerph-18-05783-f003:**
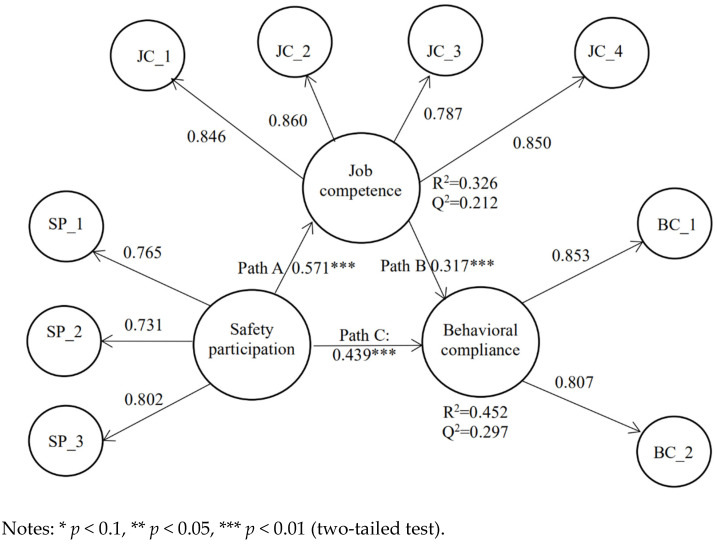
Model with job competence as a mediating variable.

**Figure 4 ijerph-18-05783-f004:**
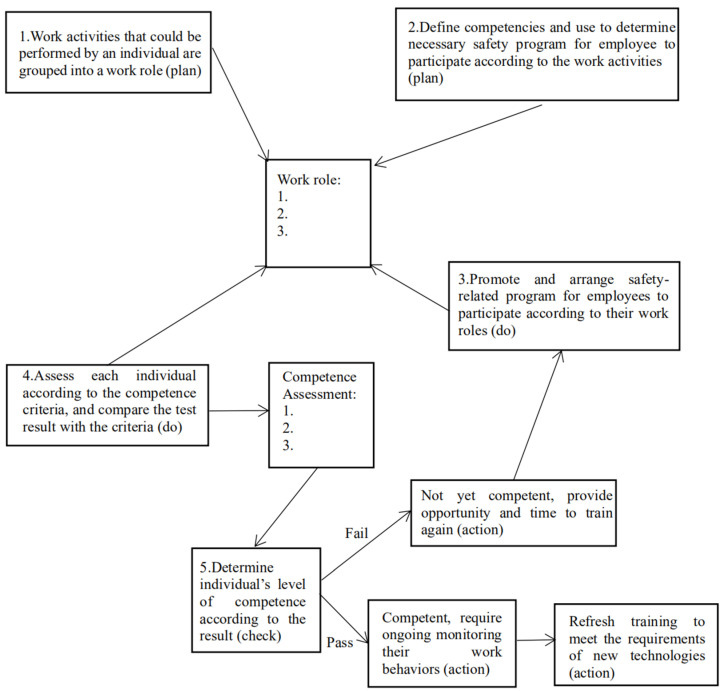
Refined management model based on this research.

**Table 1 ijerph-18-05783-t001:** Descriptive statistics of samples.

Item Description		Frequency	Percentage
Sex	Male	279	96.20%
	Female	11	3.80%
Age	≤30	46	15.90%
	31–39	215	74.10%
	40–49	22	7.60%
	≥50	7	2.40%
Department	Production Fourth Sector	134	46%
	Production Sixth Sector	156	54%
Working groups/teams	Operating team 1	76	26.20%
	Operating team 2	58	20%
	Operating team 3	81	27.90%
	Operating team 4	75	25.90%
Work position	Leader	22	7.60%
	External operator	117	40.30%
	Internal operator	151	52.10%
Years of working	≤5	12	4.20%
	6–10	224	77.20%
	11–15	24	8.30%
	≥16	30	10.30%

**Table 2 ijerph-18-05783-t002:** Survey items.

	Survey Items	Items Description
Safety Participation	SP_1	Participate in job hazard identification and risk assessment, and effectively control risks.
	SP_2	Participate in team and post safety inspection and routine inspection, and be able to find, report, and solve problems onsite in time.
	SP_3	Be able to actively participate in the emergency drill of the post and perform their own emergency responsibilities.
Behavioral Compliance	BC_1	Strictly implement all kinds of management regulations and operation procedures in daily work, without violation behavior.
	BC_2	Wearing correct personal protective equipment.
Job Competence	JC_1	Be familiar with the operation procedures and safety requirements related to the post, such as the equipment, facilities, tools and instruments of the post, understand the risks in use, maintenance and other safety requirements, be able to operate correctly and skillfully, and correctly judge and handle in case of failure.
	JC_2	Understand the risks within the territory and know the relevant risk control measures. For example, understand the types of hazardous chemicals involved in the post, be able to master the physical and chemical properties of hazardous chemicals, management requirements and emergency response measures.
	JC_3	Be familiar with the emergency procedures, responsibilities and division of work involved in the business scope of the post. For example, master the use of positive pressure air respirator, antifreeze clothing, fire clothing, gas masks, fire extinguishers and other emergency equipment.
	JC_4	Master the emergency disposal process and measures of post emergency events, emergency escape and first-aid knowledge and skills involved.

**Table 3 ijerph-18-05783-t003:** Results of validity test.

The Minimum of KMO and Bartlett’s Test		
The Kaiser–Meyer–Olkin measure of sampling adequacy		0.812
Bartlett’s test of sphericity	Approx. chi-squared	249.881
	Sig.	0.000

**Table 4 ijerph-18-05783-t004:** Measurement model results.

Constructs	Indicators	Outer Loadings	Composite Reliability	AVE
Safety participation	SP_1	0.765	0.81	0.588
	SP_2	0.731		
	SP_3	0.802		
Job competence	JC_1	0.846	0.903	0.699
	JC_2	0.86		
	JC_3	0.787		
	JC_4	0.85		
Behavioral compliance	BC_1	0.853	0.816	0.689
	BC_2	0.807		

**Table 5 ijerph-18-05783-t005:** Fornell–Larcker criterion results.

	Safety Participation	Job Competence	Behavioral Compliance
Safety participation	0.767		
Job competence	0.571	0.836	
Behavioral compliance	0.62	0.568	0.83

**Table 6 ijerph-18-05783-t006:** Cross loadings criterion results.

	Job Competence	Safety Participation	Behavioral Compliance
JC_1	0.846	0.512	0.548
JC_2	0.86	0.512	0.468
JC_3	0.787	0.395	0.388
JC_4	0.85	0.474	0.475
SP_1	0.453	0.765	0.444
SP_2	0.421	0.731	0.472
SP_3	0.438	0.802	0.507
BC_1	0.496	0.547	0.853
BC_2	0.444	0.479	0.807

**Table 7 ijerph-18-05783-t007:** Analysis of the mediating effect of job competence.

Hypothesis	Procedure	Path	Path Coefficient (β)	VAF
2, 3, 4	Indirect effect	Safety participation → Job competence → Behavioral compliance	0.181 ***	
1	Direct effect	Safety participation → Behavioral compliance	0.439 ***	31.70%
	Total effect	Indirect effect + direction effect	0.620	

Notes: * *p* < 0.1, ** *p* < 0.05, *** *p* < 0.01 (two-tailed test).

## Data Availability

We declare that our data are available anytime.

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
