# Peer review of "The Mediating Role of Job Competence between Safety Participation and Behavioral Compliance"

_ijerph, 2021, doi:10.3390/ijerph18115783_

Round 1

Reviewer 1 Report

It is important that the authors put in the Abstract, some of the conclusions obtained in this research.

The authors mention "At the construction site, the lack of safety behavioural compliance poses safety risks and a higher probability of accidents in workers", but does this situation only occur in the construction sector or does it occur in all organizations?

The authors carry out an extensive Literature Review, with many scientific articles, some of them very recent. The review addresses security in different countries, such as Nigeria, Korea, Ireland, Vietnam. This approach to the literature review gives a worldwide State of the Art knowledge.

The authors do not indicate how the initial validation of the questionnaire was done, which calls into question the entire research. Who and how was the initial validation of the questionnaires done?

The authors verified the reliability of the internal consistency of the values obtained, which allows looking at the results in a reliable way!

[Table 3] Why do the values for Indicators JC_2 and JC_4 not appear in the table? It is not clear this situation in the article...

[5.2. Practical Contribution] The authors propose a professional competence management model based on the PDCA (plan-do-check-action) management model, in line with figure 4. However, when looking at figure 4, it is not possible to understand which are the 4 PDCA phases. It would be very important that the authors clarify this situation, since the idea is very good and would be a great contribution to the scientific community.

Reviewer 2 Report

It is judged to be a very elaborately designed paper. The following points need to be considered.

First, it is necessary to check whether there is a possibility that there is an inverse causal relationship between safety participation and behavioral compliance currently set as independent and dependent variables. In addition, it is necessary to check for common method bias.

Second, it is judged to be too simple to mediate only job competence. If there are variables that are in a competitive relationship with job competence, it is necessary to perform comparative analysis between job competence and other main variables.

Third, it is necessary to address the representativeness of the sample and the problem of sample error in more detail.

Fourth, it is necessary to describe in more detail what the theoretical implications of the paper can be given.

Round 2

Reviewer 2 Report

All of things commented were revised.